# Molybdenum’s Role as an Essential Element in Enzymes Catabolizing Redox Reactions: A Review

**DOI:** 10.3390/biom14070869

**Published:** 2024-07-19

**Authors:** Jakub Piotr Adamus, Anna Ruszczyńska, Aleksandra Wyczałkowska-Tomasik

**Affiliations:** 1Faculty of Medicine, Medical University of Warsaw, 02-091 Warsaw, Poland; 2Faculty of Chemistry, Biological and Chemical Research Centre, University of Warsaw, 02-089 Warsaw, Poland; 3Department of Clinical Immunology, Medical University of Warsaw, 02-006 Warsaw, Poland

**Keywords:** molybdenum (Mo), molybdenum cofactor (MoCo), molybdenum cofactor deficiency (MoCD), xanthine oxidase (XO), aldehyde oxidase (AO), sulfite oxidase (SOX), mitochondrial amidoxime-reducing component (mARC), hepatocellular carcinoma (HCC), non-alcoholic fatty liver disease (NAFLD)

## Abstract

Molybdenum (Mo) is an essential element for human life, acting as a cofactor in various enzymes crucial for metabolic homeostasis. This review provides a comprehensive insight into the latest advances in research on molybdenum-containing enzymes and their clinical significance. One of these enzymes is xanthine oxidase (XO), which plays a pivotal role in purine catabolism, generating reactive oxygen species (ROS) capable of inducing oxidative stress and subsequent organ dysfunction. Elevated XO activity is associated with liver pathologies such as non-alcoholic fatty liver disease (NAFLD) and hepatocellular carcinoma (HCC). Aldehyde oxidases (AOs) are also molybdenum-containing enzymes that, similar to XO, participate in drug metabolism, with notable roles in the oxidation of various substrates. However, beneath its apparent efficacy, AOs’ inhibition may impact drug effectiveness and contribute to liver damage induced by hepatotoxins. Another notable molybdenum-enzyme is sulfite oxidase (SOX), which catalyzes the conversion of sulfite to sulfate, crucial for the degradation of sulfur-containing amino acids. Recent research highlights SOX’s potential as a diagnostic marker for HCC, offering promising sensitivity and specificity in distinguishing cancerous lesions. The newest member of molybdenum-containing enzymes is mitochondrial amidoxime-reducing component (mARC), involved in drug metabolism and detoxification reactions. Emerging evidence suggests its involvement in liver pathologies such as HCC and NAFLD, indicating its potential as a therapeutic target. Overall, understanding the roles of molybdenum-containing enzymes in human physiology and disease pathology is essential for advancing diagnostic and therapeutic strategies for various health conditions, particularly those related to liver dysfunction. Further research into the molecular mechanisms underlying these enzymes’ functions could lead to novel treatments and improved patient outcomes.

## 1. Introduction

Molybdenum (Mo), an element with an atomic number (Z) of 42, was first discovered in 1778 by Karl Scheele [1]. Its features closely resemble those of lead, hence the name molybdenum, which could be traced back to the Greek word *molybdos*, meaning “lead-like”. The essential role of Mo in human life was established in 1953 by De Renzo EC et al. and Richert DA et al., who identified Mo as a cofactor of xanthine oxidase (XO) [2,3]. The ubiquitous pterin-based molybdenum cofactor (MoCo) constitutes part of the active centers of all molybdenum enzymes in living organisms, without which molybdenum remains catalytically inactive.

Molybdenum’s ability to serve as a cofactor in molybdenum-containing enzymes renders it indispensable in redox reactions, where electron transfer is pivotal for biological function. During these reactions, the oxidation state of Mo alternates between IV and VI. Mo-enzymes catalyze reactions involving the transfer of two electrons to or from a substrate, which is coupled with the transfer of an oxygen atom that is either derived from or incorporated into water. Molybdenum is not directly attached to the catalytic site but its atom is complexed within a multiring organic carbon skeleton with phosphate (PO_4_^2−^) and two sulfites (S^−^) [Figure 1]. This compound, known as molybdopterin, forms the molybdenum cofactor MoCo upon coordinating with Mo.

Understanding the significance of molybdenum (Mo) in redox biology provides a framework for exploring its role in various enzymatic processes. Enzymes with different metal cofactors exhibit distinct activities, structures, and catalytic mechanisms, rooted in the properties of their respective metals.

Molybdenum enzymes operate by cycling molybdenum between its +4 and +6 oxidation states, enabling the transfer of oxygen atoms and electrons. They are integral to processes like purine metabolism, aldehyde detoxification, and sulfur amino acid metabolism. In contrast, selenium-containing enzymes have selenocysteine as their active sites. These enzymes function through redox cycles involving residue. Thus, in selenoenzymes, selenium does not serve as a cofactor. For example, in glutathione peroxidase (GPx), the selenol group (SeH) of selenocysteine is oxidized by peroxides to selenenic acid (SeOH), which is then reduced back to the selenol form by glutathione, completing the catalytic cycle. Selenium enzymes like GPx and thioredoxin reductase (TrxR) primarily function to protect cells from oxidative stress by reducing peroxides and maintaining the redox balance of proteins. These enzymes play a pivotal role in cellular defense mechanisms against oxidative damage.

Returning to the role of molybdenum in biological systems, the unique coordination chemistry of Mo in the active site provides distinct catalytic properties that are critical for various metabolic pathways involving the transfer of oxygen atoms and electrons. The number of enzymes in which Mo acts as a cofactor is finite, and this group (beyond XO) also includes sulfite oxidase (SOX), aldehyde oxidase (AO), and mitochondrial amidoxime-reducing component (mARC). Each of the aforementioned enzymes has a great contribution to *sensu lato* metabolic homeostasis. The most recently discovered of the bunch is mARC, which was isolated and identified in 2006 by Havemeyer et al. [4]. The drug-metabolizing mARC is not only able to activate N-hydroxylated prodrugs but also inactivate substances relying on subgroups’ incorporation of N-OH bonds [5]. The first molybdenum-containing oxidating enzyme—xanthine oxidase (XO)—is vital for the catalyzation of purines to uric acid [6]. Aldehyde oxidase is the primary catalyzer in the metabolism of *N*-heterocyclic compounds of both exo- and endogenous origins [7,8]. Sulfite oxidase plays an imperative role in the degradation of the amino acids methionine (Met) and cysteine (Cys) [9,10]. 

The distribution of Mo in the human body was studied by Schroeder et al. [11] via tissue analysis of 381 human cadavers. The results revealed varying levels of molybdenum across tissues, with the highest content observed in the liver (1.1 mg/kg) and kidneys (0.036 mg/kg). Considering these variations, the estimated maximum potential molybdenum stored in the body was approximately 0.13 mg/kg. Additionally, in experiments conducted by Rosoff et al. [12], the liver demonstrated the highest uptake of molybdenum (18%), followed by the kidney (9%) and pancreas (3%).

The compartmental analysis performed by Tsongas et al. [13] estimated total body molybdenum stores in healthy adults based on daily molybdenum intake ranging from 120 to 240 µg/day to be approximately 2224 µg. This estimation closely aligns with results reported by Schroeder et al. [11] (2286 µg). Moreover, the compartmental modeling employed to investigate changes in molybdenum distribution and elimination in response to varying intake levels showed urinary excretion as the primary pathway for regulating the body’s exposure to molybdenum [14].

This research highlights the organism’s capacity to adapt to molybdenum intake levels, facilitating the elimination of excess at higher intakes and conservation at lower intakes. Such an adaptive response plays a crucial role in mitigating the risks associated with both molybdenum deficiency and toxicity.

Primary nutritional molybdenum deficiency in humans is a rare phenomenon but the deficiency of molybdenum cofactor (MoCD) can occur due to genetic defects in any of the multistep enzymatic pathways synthesizing MoCo. Hence the result of present MoCD in humans is a complete loss of properly functioning XO, AO, SOX [9,15] and mARC. Among the biochemical features of MoCD is the accumulation of sulfite accompanied by a reduction in Cys; in addition, uric acid levels are significantly reduced while xanthine is elevated. The symptoms of MoCD deficiency predominantly arise due to the insufficiency of SOX, which safeguards organs, notably the brain, from the harmful effects of elevated levels of toxic sulfite [9]. The clinical presentation in these types of genetic defects is present from neonatal age and, among many symptoms, include minor dysmorphic facial features, solitary cerebral parenchymal cysts, hypoplastic pons, cerebellum, myoclonic seizures, apnea, limb hypertonia, or opisthotonos [15], and sadly, in the majority of cases, lead to the early death of the patients [16].

Even though the topic of molybdenum’s significance in human body metabolism is still being studied, we still have much to discover. This review summarizes the current state of knowledge regarding the latest advancements in research of MoCo-incorporating enzymes and their clinical importance. 

## 2. Materials and Methods

Two independent researchers searched the medical database PubMed using phrases including either full or short names of at least one of the described enzymes and/or additional terms. Phrases used in the search included “Xanthine Oxidase liver”, “Aldehyde Oxidase liver”, “XO liver pathology”, “Xanthine Oxidase Molybdenum Cofactor”, “Molybdenum Cofactor Deficiency sulphite oxidase” etc. They also reviewed references from the articles they found. In total, they collected 102 publications; of these, 91 met the inclusion criteria such as original or review publications describing the function, history, and/or biochemistry of at least one of the aforementioned enzymes. Furthermore, the articles touching on the topic of enzymes and connecting it with liver pathology were particularly useful. The publications explaining the epidemiology of liver diseases and elaborating on MoCD were also included to provide high-quality insights into the clinical aspect of liver pathophysiology. The researchers excluded studies that presented the topic of selected enzymes superficially or just briefly mentioned them in the text. 

## 3. Selected Molybdenum-Containing Oxidating Enzymes

### 3.1. Xanthine Oxidase

Xanthine oxidase (XO), also known as xanthine oxidoreductase [Figure 2], is prominently found in the epithelial cells of the intestines and the parenchymal and bile duct epithelial cells of the liver [17]. Immunohistochemical research has identified the presence of XO in the endothelial capillaries [18]. 

In vivo, XO exists in two forms: the dehydrogenase (non-superoxide-generating) form, which utilizes NAD^+^ as an electron acceptor, and the oxidase (superoxide-generating) form, which uses O_2_ as an electron acceptor. Under conditions such as ischemia and/or non-reversible proteolysis, the dehydrogenase form, prevalent under normal physiological conditions, may be transformed into the oxidative form in the majority of cells [19,20].

Xanthine oxidase is a homodimer with a molecular weight of 270 kDa. Each monomer contains a molybdenum center where substrate hydroxylation occurs, a flavin adenine dinucleotide (FAD) cofactor facilitating electron transfer from the molybdenum center, and two iron–sulfur centers ([2Fe-2S] clusters. The redox reaction centers are almost linearly positioned in the order of molybdopterin, [2Fe-2S] centers, and FAD. One of the Fe-S centers has a higher redox potential. Molybdenum binds with the pterin ring through a sulfur atom, with a further sulfur atom and two oxygen atoms coordinated to the molybdenum and exposed to solvent. One of the oxygen atoms is derived from a water molecule and incorporated into the substrate (hypoxanthine and xanthine). The enzyme is reduced and receives H^+^ + 2e^−^ from the substrate, reducing the molybdenum center from Mo(VI) to Mo(IV). This is followed by electron transfer through the iron–sulfur clusters to the FAD cofactor, ultimately releasing reducing equivalents. The final electron acceptor, which could be NAD^+^ or oxygen molecule, is reduced [21,22,23,24,25]. 

**Figure 2 biomolecules-14-00869-f002:**
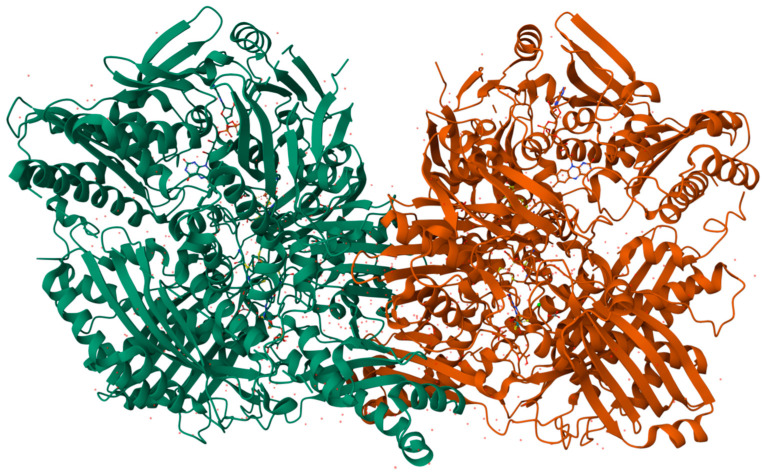
3D structure of bovine xanthine oxidase (XO), protein cleaved form (Animated demonstration in the Appendix A) [26].

Xanthine oxidase catalyzes the final two steps in purine catabolism, using hypoxanthine and subsequently xanthine as substrates for uric acid synthesis [6] [Figure 3]. The reactions are as follows:Hypoxanthine + H_2_O + O_2_ ↔ Xanthine + H_2_O_2_
Xanthine + H_2_O + O_2_ ↔ Uric acid + H_2_O_2_

As byproducts of these reactions, XO produces superoxide anion (O_2_^−^) and hydrogen peroxide (H_2_O_2_). The first one is an essential precursor for peroxynitrate radical (ONOO**^•^**) and hydroxide radical (OH**^•^**) [6,17,27]. Some reactive oxygen species (ROS) generated by these byproducts can mediate oxidative stress and organ dysfunction [27,28]. These ROS influence cell function through DNA and protein fragmentation. They can also disturb the continuity of the cell membrane by peroxidation of the membrane lipids and stress host tissue directly, leading to irreversible damage [29]. These ROS characteristics are vital for defense against some infections, among which malaria is worth highlighting due to its high prevalence, which reached around 247 million cases globally in 2021 alone [30]. 

In this disease, especially of *Plasmodium falciparum* etiology, hepatocellular dysfunction with markedly elevated levels of XO, uric acid (XO metabolic product), liver function enzymes (GOT and GTP), catalase as well as bilirubin levels are reported by Iwalokun et al. [31]. In addition, increased XO activity with elevated transaminase levels was found to indicate the presence of liver disease [32]. Normally, the levels of XO and liver function enzymes in the circulation are low. However, their increased levels are greatly correlated with liver pathologies. Liver enzyme levels can drastically increase in the serum when hepatocytes undergo lysis [33], which may indicate a relationship between hepatocyte lysis and the release of XO into the bloodstream [31]. 

The serum level of xanthine oxidase varies and depends on the primary liver disease. The study conducted by Batelli et al. on the 64-patient group with chronic liver disease concluded that sick patients had significantly higher serum XO levels compared to the 12-people control group. The greatest increase of said oxidase was noted in cases of cholestatic disorders; some elevation was present in chronic hepatitis patients, but not in cirrhosis [32]. An evident increase in serum XO levels in individuals with chronic liver disease appears to indicate the presence of cholestasis. 

The other disease that exhibits an indirect correlation with XO activity is non-alcoholic fatty liver disease (NAFLD). NAFLD has a high prevalence of 47.8% in the United States [34,35] and 26.9% in Europe [34,36]. It is one of the most common chronic liver diseases globally [37]. High serum uric acid levels are widespread metabolic abnormalities present in obese patients [38] who are at great risk of NAFLD [39]. Chengfu Xu et al. studied the relationship between NAFLD and hyperuricemia (uric acid as the product of xanthine’s oxidation). They firmly linked NAFLD with the subsequent onset of hyperuricemia. Furthermore, they also discovered elevated expression and activity of XO in cellular and mouse models of NAFLD. This increase might elucidate the molecular connection between NAFLD and high serum uric acid levels. Moreover, their findings demonstrate that XO plays a pivotal role in regulating NAFLD [40] and could potentially serve as an innovative therapeutic target for patients afflicted with this condition. 

### 3.2. Aldehyde Oxidase

The first references in literature to the xanthine oxidase-resembling enzyme, aldehyde oxidase, date back to the 1930s [41] and 1940s [42]. Due to the increasingly frequent association of AO with the principal metabolic pathways of drugs [43], there has been a visible surge in interest within the scientific community, leading to numerous studies and articles published on AO. This interest peaked, according to PubMed, with 132 articles on AO published in 2018 alone [44]. 

Aldehyde oxidase (AO) [Figure 4] is an enzyme homologous to xanthine oxidase, and just like XO, is a complex molybdoflavoprotein. Moreover, both oxidases demonstrate a notable level of similarity in their respective amino-acid sequences [45] and require the same cofactors. Each identical subunit of the AO homodimer is about 140–150 kDa when active. The single subunit can be further divided into three separate domains: the smallest *N*-terminal 20 kDa domain responsible for binding the two non-identical iron-containing aggregates, the central 40 kDa domain contains a binding site for a flavin adenine dinucleotide (FAD), and the largest C-terminal 85 kDa domain contains MoCo and a substrate-binding site in close proximity [46,47]. Even though there are great resemblances between XO and AO functions, for example, both enzymes enable oxidation as well as reduction reactions across a broad spectrum of substrates, with oxidation reactions being significantly more prevalent [45], significant differences also exist. Remarkably, there are distinctions in inhibitor and substrate specificities between XO and AO [48]. The only electron acceptor for AO is molecular oxygen [49]. AO can oxidase a broader range of substrates compared to XO [49,50]. Compounds with an aldehyde group, *N*-heterocycles, or nitro compounds are some examples of AO substrates [45,51]. The mechanism of AO-catalyzed oxidation is as follows: the substrate undergoes oxidation to produce the product at the MoCo. Subsequently, the reducing equivalents are transferred to FAD, which is then reoxidized by the molecular oxygen. The iron-containing centers play a role as mediators in electron transfer between MoCo and the flavin cofactor. Additionally, they act as electron sinks, storing reducing equivalents throughout the catalytic process [7,8,45].

AO-mediated clearance of drugs predominantly takes place in the liver. The liver exhibits the greatest AO activity [44] but the degree of AO activity varies between species [53]. Moriwaki et al. [54] reported that outside the liver, other tissues and organs have noticeable AO activity [50,54], including endocrine, respiratory, digestive, and kidney. In addition, they present cell-specific localization. For instance, high AO activity was present in the renal tissue, especially in proximal and distal convoluted tubes and collecting ducts. The respiratory tract’s epithelium had abundant AO after immunohistochemical staining [54]. 

### 3.3. Involvement in Drug Metabolism of Aldehyde Oxidase and Xanthine Oxidase

Aldehyde oxidase is an enzyme that has an important role in drug metabolism in the liver [55]. A study conducted in 2013 by Cexiong Fu et al. found a significant drop in aldehyde oxidase 1 (AO1) levels in human liver cytosols of donors with chronic alcohol consumption compared to controls, although interestingly, the cells preserved almost all AO1 expression [56].

Scott Obach et al. conducted the biggest study at that time on human liver-derived AO interactions with 239 drugs. The results show that as many as 36 (out of 239) frequently administered drugs led to AO inhibition at a level greater than 80%. This distinctive group was subsequently subjected to further investigation to determine their IC_50_ values. Raloxifene, a selective estrogen receptor modulator, demonstrated the greatest potency as an inhibitory agent (IC_50_ = 2.9 nM), with tamoxifen, estradiol, and ethinyl estradiol also showing notable inhibitory effects [43].

In 2014, Choughule et al. reported the vital function of AO and XO in the metabolism of 6-mercaptopurine (6MP) [57], an agent administered for the treatment of childhood acute lymphoblastic leukemia [58]. Oxidation and methylation of 6MP produce inactive metabolites. The roles of AO and XO were investigated via the utilization of specific inhibitors—raloxifene and febuxostat. This study established that both AO and XO participate in 6MP oxidation to 6-thixanthine (6TX) intermediate, while only XO is involved in the conversion of 6TX to 6-thiouric acid (6TUA). A combined therapy consisting of an XO inhibitor with 6MP has been shown to increase the bioavailability of 6MP [58] [Figure 5]. 

According to Shakir Ali et al., these oxidases, apart from drug–drug interactions, contribute to the hepatic damage inflicted by free radicals thanks to the accumulation of a variety of hepatotoxins agents such as carbon tetrachloride (CCl_4_), chloroform (CHCl_3_), and thioacetamide (TAA). Elevated levels of CCl_4_, CHCl_3_, and TAA resulted in increased levels of molybdoproteins. Interestingly, liver damage caused by glutathione-depleting substances did not lead to a rise in molybdenum-containing oxidases, hence they did not participate in amplification of hepatic damage [59].

These studies have underlined the meaningful role of AO and XO in drug metabolism and drug–drug interactions within the liver. It suggests the clinical relevance of understanding AO-/XO-mediated drug interactions with liver physiology.

### 3.4. Sulfite Oxidase 

The properties of the human SOX molecule were investigated by Johnson et al. in 1976 [60]. Those researchers estimated the weight of the SOX molecule [Figure 6] to be approximately 61.1 kDa. They also discovered that human SOX is more negatively charged compared to the SOX isolated from rat liver [60]. Sulfite oxidase is widely regarded as the most crucial molybdenum-containing oxidating enzyme for human health [15] since it catalyzes the final stage in oxidative degradation of sulfur-containing amino acids (e.g., cysteine) and lipids. SOX facilitates the conversion of sulfite into sulfate. SOX is situated in the intermembrane space of mitochondria and shuttles electrons from sulfite oxidation to cytochrome c (Cyt *c*), hence connecting sulfite oxidation to the reduction of Cyt *c* [9,10,15]. 

The animal dimeric structure of the SOX consists of one molybdenum domain and one cytochrome *b*_5_-type heme domain [60,62]. The latter is an electron acceptor from the molybdenum center. The catalytic process of sulfite oxidase encompasses the oxidation of sulfite coupled with the reduction of molybdenum, succeeded by two distinct electron transfer phases via the cytochrome *b*_5_ domain to Cyt *c*. This process is characterized by significant spatial movements of the heme domain within sulfite oxidase [10]. SOX and similar enzymes could be present in bacteria [63] and plants [64] as well. The form in microorganisms is assembled into a heterodimer consisting of a single subunit containing molybdenum and Cyt *c*, while in the latter, a homodimer composed of two Mo subunits without the heme domain [63,64].

### 3.5. Sulfite Oxidase in Modern Clinical Use

Recent clinical studies have begun a new chapter for possible clinical use of SOX activity such as in the diagnosis of hepatocellular carcinoma (HCC) [65]. According to the World Health Organization’s estimates for 2022, 760,000 people died of liver cancer making it the third leading cause of cancer-related deaths wordlwide [66]. Eastern Asia and sub-Saharan Africa are regions that suffer the most from HCC. This cancer follows a similar high prevalence pattern of chronic hepatitis B virus (HBV) and approximately 80% of HCC cases occur there [67]. There are numerous etiologies of HCC. Apart from HBV infection, chronic hepatitis C virus (HCV) infection, or non-alcoholic steatohepatitis (NASH), a more severe form of NAFLD characterized by concurrent inflammation. All of the above can lead to cirrhosis and eventually the development of HCC [68,69]. 

The high global prevalence of HCC and the difficult diagnostic process poses a significant challenge for modern medicine. HCC markers such as heat shock protein 70, glypican 3, and glutathione synthase have roles in the cancer’s diagnostic process [70,71] but have relatively low sensitivity in differentiating HCCs, hence the need for improvements in this area. 

In 2010, Satow et al. reported an elevation of aldo-keto reductase family 1 member B10 (AKR1B10) in HCC [72]. The following year, Guang-Zhi Jin et al. stated that the SOX could be a suitable immunohistochemical marker for distinguishing well-differentiated small HCC (WD-sHCC) from high-grade dysplastic nodules (HGDNs) [73], which are precancerous lesions with a high risk of malignant transformation [74,75]. Jin GZ. et al. later combined previous research on HCC [70,71,72,73] and conducted a pioneering study establishing that a marker combination that includes SOX is a meaningful contributor to immunopathological diagnosis in HCC cases when distinguishing WD-sHCC from HGDNs. These researchers found that the combination of markers SOX + AKR1B10 + CD34 yielded promising sensitivity (93.8%) and specificity (95.2%) in the differentiation of WD-sHCC from HGDNs [65]. 

SOX is a vital biochemical component not only for the proper function of the liver but also for brain physiology. The manifestations of molybdenum cofactor deficiency primarily stem from the inadequate presence of SOX, which stands as the protector of organs, particularly the brain, against the detrimental impact of increased levels of toxic sulfite. The MoCD also leads to sulfite accumulation [9], which has a detrimental effect on neurons. Aggregated sulfite in serum and plasma crosses the blood–brain barrier and leads to neuron death [10], diminished ATP synthesis [76], and (indirectly) stimulation of glutamate receptors [77]. The latter might be the underlying reason for neural symptoms associated with MoCD, such as convulsions or seizures, leading to irreversible neuronal damage visible as white matter loss [16]. Recent studies performed on animal models (rats) show great connections between SOX levels and central nervous system (CNS) function. In 2012, Kocamaz et al. reported that sulfite accumulation led to a significant drop in the total number of pyramidal neurons in the hippocampus [78]. Cells that demonstrate high SOX expression in CNS are astrocytes. One of the latest studies on SOX shows that SOX gene knockdown or replacement of Mo with tungsten (W) in MoCo decreases NO synthesis by the glia during hypoxia [79].

### 3.6. Mitochondrial Amidoxime-Reducing Component 

Molybdenum, playing a crucial role in both oxidation and reduction processes, is also a key element in the most recently discovered human molybdoenzyme—mitochondrial amidoxime-reducing component (mARC)—which was identified and isolated in 2006 [4]. mARC, alongside heme-containing cytochrome *b*_5_ (Cyt *b*_5_) and NADH-dependent FAD/cytochrome b_5_ reductase (Cyt *b*_5_R), form a three-enzyme complex localized in the outer mitochondrial membrane [4,80,81]. Interestingly, mARC contains MoCo, which is homologous to the domain of molybdenum cofactor sulfurase [82]. In all mammalian genomes examined up until now, there are two mARC genes: MTARC1 and MTARC2. In humans, these genes are localized on chromosome 1 and encode two proteins, mARC1 [Figure 7] and mARC2 [81,82,83,84], respectively. 

mARC, with versatile capabilities, participates in the reduction of various substrates. Its initial discovery was linked to the reduction of benzamidoxime [86]. Furthermore, mARC is involved in reducing compounds such as *N*-hydroxy-valdecoxib and *N*-hydroxy-benzenesulfonamide, the members of a family of *N*-hydroxy sulfonamides [87]. *N*-hydroxamic acids, such as benzhydroxamic acid and bufexamac [88] as well as *N*-hydroxyguanidines like *N^ω^*-hydroxy-L-arginine [89] also undergo reduction via mARC and play essential roles in various biochemical pathways. Figure 8 presents the selected molecular structures of the mARC substrates. 

For compound reduction to occur, mARC requires electrons; NADH is the electron donor. The subatomic particle then passes through the FAD-containing Cyt *b*_5_R and Cyt *b*_5_, eventually reaching the mARC protein, where the substrate is reduced in the mARC molybdenum-active site [82]. mARC is present in great quantities in the kidneys and the liver and thus actively participates in the detoxification of *N*-hydroxylated substrates [1]. 

The mitochondrial amidoxime-reducing component serves as an enzyme that metabolizes drugs, with the ability to activate N-hydroxylated prodrugs. On the other hand, mARC can also deactivate drug substances that depend on functional groups with N-OH bonds [5]. The cytostatic agent N-hydroxyurea, which inhibits ribonucleotide reductase, is employed in the treatment of sickle-cell disease and certain types of cancers and serves as an excellent substrate for mARC1 [90]. In a related context, Zhang et al. found that hydroxamic acids are utilized in pharmacophores that target metalloproteins, a principle that is applied to the inhibitors of zinc-containing histone deacetylase [91]. 

The role of mARC in detoxification reactions is an important point of discussion. Following the identification of mammalian mARC [4], a new mARC discovery emerged. In 2008, Kozmin et al. found that two homologs of mARC, namely YcbX and YiiM, aid in bolstering the resistance of *Escherichia coli* bacteria to 6-hydroxylaminopurine [92]. Protecting cells from the harmful effects of mutagenic N-hydroxylated nucleobases and nucleotides is one of the other roles of mARC proteins [93]. mARCs have also been observed to reduce said substances [94]. However, the relationship between the functional groups, which have been shown to be reduced by mARC, and the physiological function of the enzymes remains uncertain [5].

### 3.7. mARC as a Significant Component in Human Disease

Increasingly, evidence suggests that mARC enzymes may play roles in physiological processes as well as contribute to human disease. Therefore, it seems that the function of mARC is not solely restricted to xenobiotic metabolism. 

Levels of mARC, similar to SOX, might show a correlation with the presence of HCC in patients’ livers. A compelling example was investigated in 2020 by Wu et al., who confirmed the involvement of mARC in HCC. Their findings revealed that mARC2 can hinder the progression of HCC by competing with the tumor suppressor protein p27 for degradation via the same ubiquitin E3 ligase, RNF123 (also known as KPC1) [95]. Reduced mARC2 expression stood out as an independent risk factor for a poor prognosis. Hence, it was notably linked to clinicopathological features of HCC, including AFP levels, tumor grade, and tumor size [95]. In the follow-up study, also performed by Wu et al., uncovered a negative relationship between the expression levels of MTARC2, Cyt *b*_5_, and Cyt *b*_5_R, as well as HCC tumor size, metastasis risk, and progression. This led to the proposal that the expression levels of MTARC2 and its associated electron carrier proteins could serve as a prognostic indicator in HCC patients [96].

mARC involvement in liver pathologies does not end with HCC. The activity of mARC exhibits a correlation with NAFLD and NASH. The recent genome-wide association study (GWAS) conducted by Emdin et al. discovered that the human variant mARC1-p.A165T seemed to offer protection against liver cirrhosis. Additionally, it was associated with reduced liver fat, blood cholesterol levels, and circulating liver enzymes [97]. In Addition, many other GWAS also confirmed a connection between liver disease and mARC1 [98,99,100]. According to Friedman et al., there are currently no pharmacotherapeutic alternatives for the treatment and prevention of NAFLD and NASH [101]. Therefore, mARC1 could be considered a new drug target, for example in patients suffering from obesity [102].

## 4. Conclusions

In conclusion, molybdenum (Mo) stands as a fundamental element in human metabolism, serving as a cofactor in enzymes vital for various physiological processes. Recognized for its essentiality since the 1950s, research has delved into understanding Mo’s distribution in tissues, metabolic functions, and clinical implications.

Predominantly found in organs like the liver, kidneys, and blood, Mo deficiency, particularly in molybdenum cofactor (MoCo) synthesis pathways, can lead to severe conditions such as MoCo deficiency (MoCD), impacting the proper functioning of MoCo-dependent enzymes like xanthine oxidase (XO), sulfite oxidase (SOX), aldehyde oxidase (AO), and mitochondrial amidoxime-reducing component (mARC).

Recent advancements in research have highlighted the clinical importance of MoCo-incorporating enzymes, shedding light on their roles in drug metabolism, liver physiology, and disease pathogenesis. XO and AO play pivotal roles in drug metabolism, particularly in the liver, with implications for pharmacotherapy optimization and management of liver-related conditions.

SOX, critical for sulfur-containing amino acid degradation and brain protection, shows promise as a diagnostic marker for hepatocellular carcinoma (HCC) and holds implications for neurodegenerative disorders. Additionally, mARC’s versatile capabilities in detoxification reactions and its involvement in liver pathologies underscore its significance as a potential prognostic indicator and therapeutic target.

Further exploration into the mechanisms of MoCo-incorporating enzymes and their implications for human health and disease is essential. By deepening our understanding of Mo’s roles, we may unlock novel diagnostic and therapeutic strategies, ultimately advancing patient care and medical knowledge.

## Figures and Tables

**Figure 1 biomolecules-14-00869-f001:**
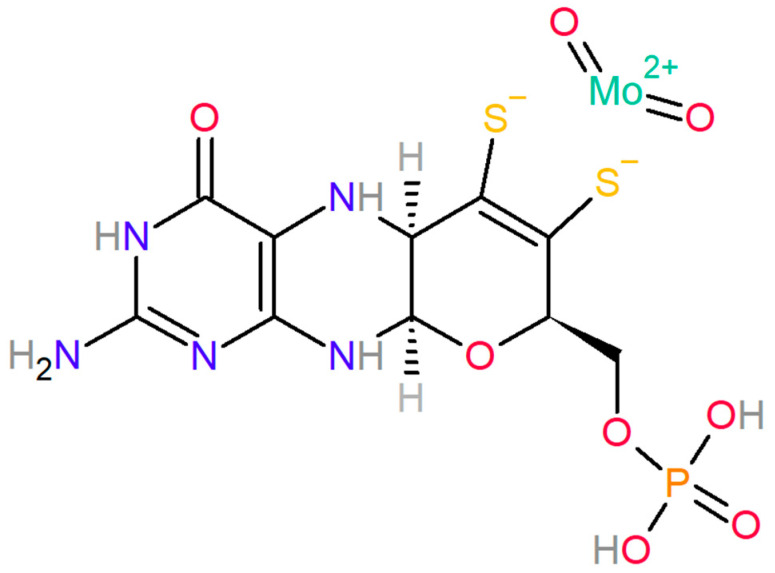
**Molybdenum cofactor (MoCo)** molecule in 2D projection. The colors and corresponding atoms are as follows: yellow (sulfur; S) dark grey (carbon; C), navy (nitrogen; N), red (oxygen; O), light grey (hydrogen; H), orange (phosphorus; P), turquoise (molybdenum; Mo). The aforementioned colors are used throughout the remaining 2D figures.

**Figure 3 biomolecules-14-00869-f003:**
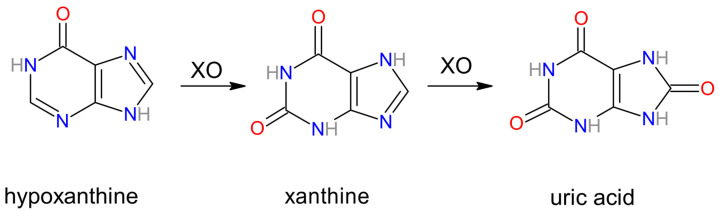
Selected reactions catalyzed by xanthine oxidase (XO), with pictures of hypoxanthine, xanthine, and uric acid molecules.

**Figure 4 biomolecules-14-00869-f004:**
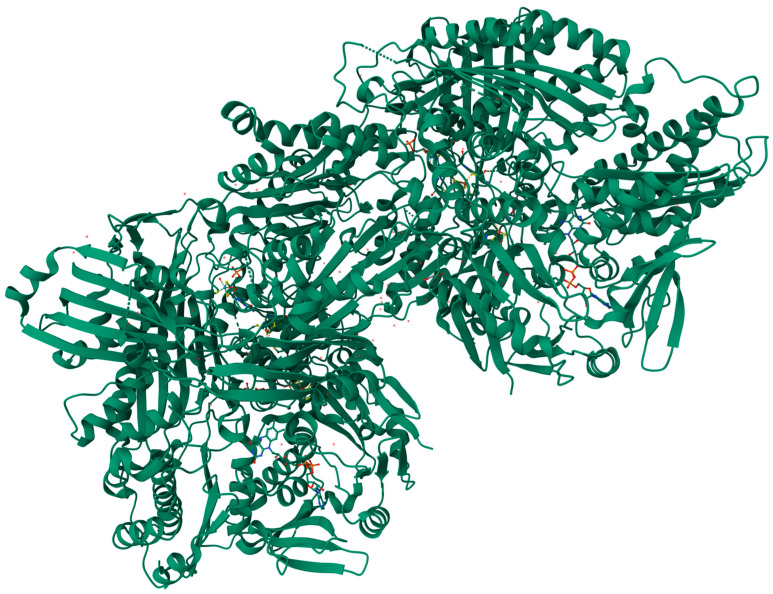
3D structure of human aldehyde oxidase (AO) protein (Animated demonstration in the Appendix A) [52].

**Figure 5 biomolecules-14-00869-f005:**
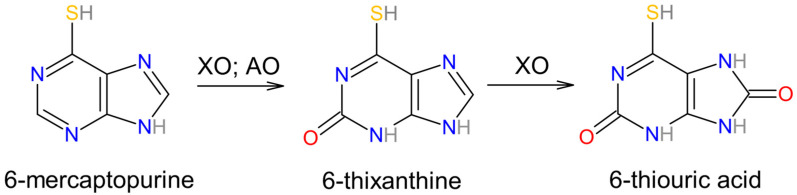
6-mercaptopurine (6MP) conversion into 6-thixanthine (6TX) intermediate, which is transformed into 6-thiouric acid (6TUA).

**Figure 6 biomolecules-14-00869-f006:**
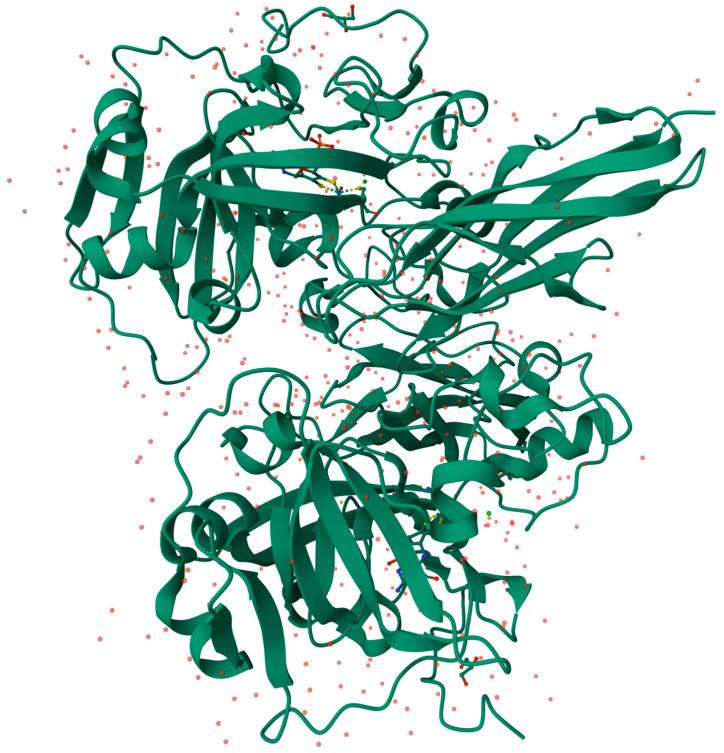
3D structure of recombinant chicken sulfite oxidase (SOX) protein at resting state (Animated demonstration in the Appendix A) [61].

**Figure 7 biomolecules-14-00869-f007:**
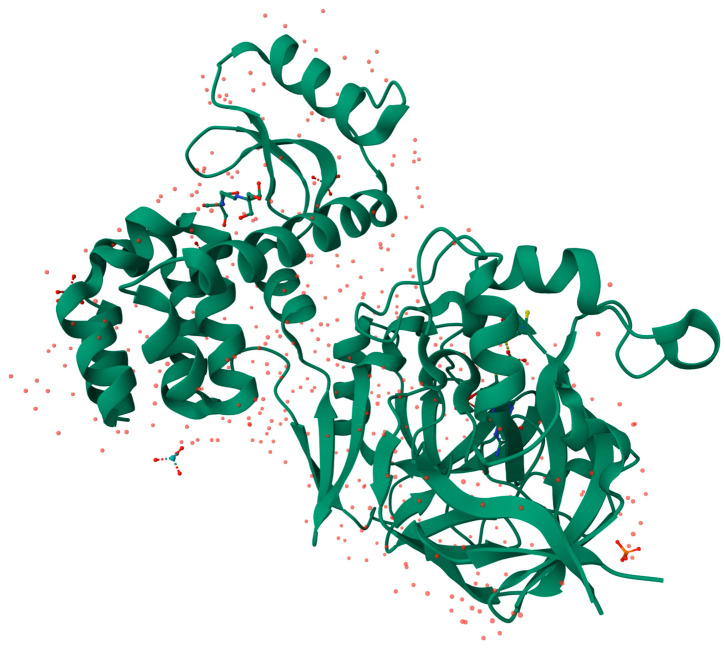
3D structure of mitochondrial amidoxime-reducing component 1 (mARC1) protein (Animated demonstration in the Appendix A) [85].

**Figure 8 biomolecules-14-00869-f008:**
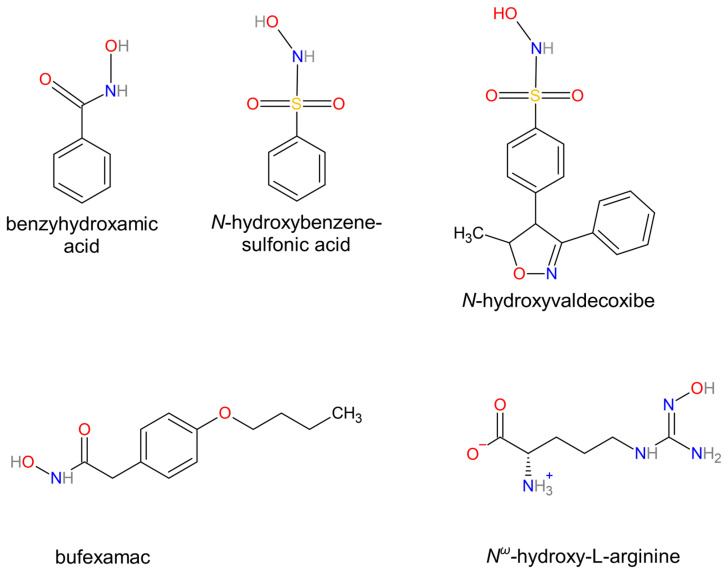
Selected mARC substrates.

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
