# Peer review of "Molybdenum’s Role as an Essential Element in Enzymes Catabolizing Redox Reactions: A Review"

_biomolecules, 2024, doi:10.3390/biom14070869_

Round 1

Reviewer 1 Report

Comments and Suggestions for Authors

This is a nice review and summary on redox enzymes using Molybdenum as the cofactor.  The manuscript is well-written and up-to-date. I have two comments:

1): the authors provide in might elaborate more on the mechanism by which Molybdenum functions in eletron transfer and redox reactions. The authors can use one protein (e.g., XO) as an example.

2): there are redox enzymes using as other metals (such as Selenum) as cofactors. A comparison of Mo-containing redox enzymes with other redox enzymes is welcome.

Author Response

Thank you very much for your valuable comments. We reviewed the text in light of your suggestions, also keeping in mind the need to maintain a concise length and stick to the main topic. The subject is complex and challenging, so we indeed consulted a significant amount of new literature, which was informative and enriched our understanding of the issue.

Comment 1: The authors might elaborate more on the mechanism by which Molybdenum functions in electron transfer and redox reactions. The authors can use one protein (e.g., XO) as an example.

Response: We have revised the manuscript to include a detailed explanation of the mechanism by which molybdenum functions in electron transfer and redox reactions, using xanthine oxidase (XO) as an example. Specifically, we have elaborated on the cycling of molybdenum between its +4 and +6 oxidation states and how this enables the transfer of oxygen atoms and electrons. The revised section can be found on page 4, line 146-164:

In vivo, XO exists in two forms: the dehydrogenase (non-superoxide-generating) form, which utilizes NAD+ as an electron acceptor, and the oxidase (superox-ide-generating) form, which uses O2 as an electron acceptor. Under conditions such as ischemia and/or non-reversible proteolysis, the dehydrogenase form, prevalent under normal physiological conditions prevalent under normal conditions, may be trans-formed into oxidative form in the majority of the cells [18, 19].

Xanthine oxidase is a homodimer with a molecular weight of 270 kDa. Each monomer contains a molybdenum center where substrate hydroxylation occurs, a fla-vin adenine dinucleotide (FAD) cofactor facilitating electron transfer from the molyb-denum center, and two iron-sulfur centers ([2Fe-2S] clusters). The redox reaction cen-ters are almost linearly positioned in the order of molybdopterin, [2Fe-2S] centers and FAD. One of the Fe-S centres has a higher redox potential. Molybdenum binds with the pterin ring through a sulfur atom, with a further sulfur atom and two oxygen atoms coordinated to the molybdenum and exposed to solvent. One of the oxygen atoms is derived from a water molecule and incorporated into the substrate (hypoxanthine and xanthine). The enzyme is reduced and receives H+ + 2e- from the substrate, reducing the molybdenum center from Mo(VI) to Mo(IV). This is followed by electron transfer through the iron-sulfur clusters to the FAD cofactor, ultimately releasing reducing equivalents. The final electron acceptor, which could be NAD+ or oxygen molecule, is reduced [20-24].

Comment 2: There are redox enzymes using other metals (such as Selenium) as cofactors. A comparison of Mo-containing redox enzymes with other redox enzymes is welcome.

Response: While we recognize the importance of comparing molybdenum-containing redox enzymes with other metal-containing redox enzymes, such as selenium-containing enzymes, we also acknowledge that this review is already quite extensive. Therefore, we have provided a brief comparison to address the differences in structure, catalytic mechanisms, and function, without going into exhaustive detail. This approach ensures that we maintain focus on the primary topic while acknowledging the broader context. The revised section can be found on page 2, line 61-78:

Understanding the significance of molybdenum (Mo) in redox biology provides a framework for exploring its role in various enzymatic processes. Enzymes with differ-ent metal cofactors exhibit distinct activities, structures, and catalytic mechanisms, rooted in the properties of their respective metals.

Molybdenum enzymes operate by cycling molybdenum between its +4 and +6 ox-idation states, enabling the transfer of oxygen atoms and electrons. They are integral to processes like purine metabolism, aldehyde detoxification, and sulfur amino acid me-tabolism. In contrast, selenium-containing enzymes have selenocysteine as their active sites. These enzymes function through redox cycles involving residue. Thus, in seleno-enzymes, selenium does not serve as a cofactor. For example, in glutathione peroxidase (GPx), the selenol group (SeH) of selenocysteine is oxidized by peroxides to selenenic acid (SeOH), which is then reduced back to the selenol form by glutathione, completing the catalytic cycle. Selenium enzymes like GPx and thioredoxin reductase (TrxR) pri-marily function to protect cells from oxidative stress by reducing peroxides and main-taining the redox balance of proteins. These enzymes play a pivotal role in cellular de-fense mechanisms against oxidative damage.

Reviewer 2 Report

Comments and Suggestions for Authors

The authors present a work focused on molybdenum-based enzymes and their role in biomedicine. 

The authors should improve the figures quality, protein structure, including the PDB reference in the figure caption.

It is highly recommended to include a detailed description of the reaction mechanism, images of the active center, and conformational changes of the enzyme to make the manuscript more comprehensible and enganging.

Author Response

Thank you for your feedback. We have carefully considered your comments and made several improvements to enhance the quality and comprehensibility of our manuscript. Below are our responses to your specific suggestions:

  1. Figures Quality and Protein Structure:

We have improved the quality of all drawings, ensuring they are now in a consistent style, higher resolution, and more readable. The references are also included in the captions where necessary. We have also included the PDB references in the figure captions for better context and to provide a direct link to the structural data.

  1. Detailed Description of Reaction Mechanism:

Additionally, we have enhanced the manuscript by providing a more detailed description of the reaction mechanism of molybdenum-containing enzymes. We have specifically elaborated on the electron transfer and redox reactions using xanthine oxidase (XO) as an example hoping that this addition provides a clearer understanding of how molybdenum functions in these processes. We  hope that these changes provide a clearer understanding of how molybdenum’s coordination within the active site facilitates the catalytic process and how conformational shifts enable substrate binding and product release. The revised section can be found on page 2, line: 48-54 and page 4 line 146-164:

In vivo, XO exists in two forms: the dehydrogenase (non-superoxide-generating) form, which utilizes NAD+ as an electron acceptor, and the oxidase (superox-ide-generating) form, which uses O2 as an electron acceptor. Under conditions such as ischemia and/or non-reversible proteolysis, the dehydrogenase form, prevalent under normal physiological conditions prevalent under normal conditions, may be trans-formed into oxidative form in the majority of the cells [18, 19].

and page 4 line 146-164:

Xanthine oxidase is a homodimer with a molecular weight of 270 kDa. Each monomer contains a molybdenum center where substrate hydroxylation occurs, a flavin adenine dinucleotide (FAD) cofactor facilitating electron transfer from the molyb-denum center, and two iron-sulfur centers ([2Fe-2S] clusters). The redox reaction centers are almost linearly positioned in the order of molybdopterin, [2Fe-2S] centers and FAD. One of the Fe-S centres has a higher redox potential. Molybdenum binds with the pterin ring through a sulfur atom, with a further sulfur atom and two oxygen atoms coordinated to the molybdenum and exposed to solvent. One of the oxygen atoms is derived from a water molecule and incorporated into the substrate (hypoxanthine and xanthine). The enzyme is reduced and receives H+ + 2e- from the substrate, reducing the molybdenum center from Mo(VI) to Mo(IV). This is followed by electron transfer through the iron-sulfur clusters to the FAD cofactor, ultimately releasing reducing equivalents. The final electron acceptor, which could be NAD+ or oxygen molecule, is reduced [20-24].